# Metagenomics Approach to the Intestinal Microbiome Structure and Abundance in High-Fat-Diet-Induced Hyperlipidemic Rat Fed with (−)-Epigallocatechin-3-Gallate Nanoparticles

**DOI:** 10.3390/molecules27154894

**Published:** 2022-07-31

**Authors:** Zhiyin Chen, Baogui Liu, Zhihua Gong, Hua Huang, Yihui Gong, Wenjun Xiao

**Affiliations:** 1Key Lab of Tea Science of Ministry of Education, Hunan Agricultural University, Changsha 410128, China; chenzhiyin100@163.com (Z.C.); liubaogui6518@163.com (B.L.); gzh041211@163.com (Z.G.); 2College of Agriculture & Biotechnology, Hunan University of Humanities, Science & Technology, Loudi 417000, China; gyhzgh@163.com; 3National Research Center of Engineering Technology for Utilization of Botanical Functional Ingredients, Hunan Agricultural University, Changsha 410128, China; 4Co-Innovation Center of Education Ministry for Utilization of Botanical Functional Ingredients, Hunan Agricultural University, Changsha 410128, China; 5Key Laboratory for Evaluation and Utilization of Gene Resources of Horticultural Crops, Ministry of Agriculture and Rural Affairs of China, Changsha 410128, China; 6Key Laboratory of South Subtropical Fruit Biology and Genetic Resource Utilization, Minis-Try of Agriculture and Rural Affairs, Guang-Dong Provincial Key Laboratory of Tropical and Subtropical Fruit Tree Research, Institute of Fruit Tree Re-Search, Guangdong Academy of Agricultural Sciences, Guangzhou 510640, China; helly.blog.com@163.com

**Keywords:** (−)-epigallocatechin-3-gallate, nanoparticles, intestinal microbiome, hyperlipidemia, metagenomics

## Abstract

The effects of nanoparticles (NPs) on microbiota homeostasis and their physiological relevance are still unclear. Herein, we compared the modulation and consequent pharmacological effects of oral administration of (−)-epigallocatechin-3-gallate (EGCG)-loaded β-cyclodextrin (β-CD) NPs (EGCG@β-CD NPs) and EGCG on gut microbiota. EGCG@β-CD NPs were prepared using self-assembly and their influence on the intestinal microbiome structure was analyzed using a metagenomics approach. The “Encapsulation efficiency (EE), particle size, polydispersity index (PDI), zeta potential” of EGCG@β-CD NPs were recorded as 98.27 ± 0.36%, 124.6 nm, 0.313 and –24.3 mV, respectively. Surface morphology of EGCG@β-CD NPs was observed as spherical. Fourier-transform infrared spectroscopy (FT-IR), X-ray diffraction (XRD) and molecular docking studies confirmed that EGCG could be well encapsulated in β-CD and formed as EGCG@β-CD NPs. After being continuously administered EGCG@β-CD NPs for 8 weeks, the serum cholesterol (TC), low-density lipoprotein cholesterol (LDL-C) and liver malondialdehyde (MDA) levels in the rats were significantly decreased, while the levels of catalase (CAT) and apolipoprotein-A1 (apo-A1) in the liver increased significantly in the hyperlipidemia model of rats, when compared to the high-fat-diet group. Furthermore, metagenomic analysis revealed that the ratio of Verrucomicrobia/Bacteroidetes was altered and Bacteroidetes decreased in the high-fat diet +200 mg/kg·bw EGCG@β-CD NPs group, while the abundance of Verrucomicrobia was significantly increased, especially *Akkermansia muciniphila* in rat feces. EGCG@β-CD NPs could be a promising EGCG delivery strategy to modulate the gut microbiota, enhancing its employment in the prevention of hyperlipidemia.

## 1. Introduction

Nowadays, accumulating and emerging studies have demonstrated that nanoparticles, such as liposomes [1], polymeric nanoparticles [2], solid lipid nanoparticles [3] and lipid nanoparticles [4], can become effective targeted delivery vehicles. Nanoparticles are nano-sized (10–1000 nm) particles that enhance the solubility of hydrophobic molecules and bioavailability of active substances, control the release of therapeutic agents and reduce drug toxicity. Nanoparticle drug delivery systems are governed by two components: the drug and a carrier, selected according to the properties of the drug and the desired therapeutic effect. Drugs generally have the characteristics of poor solubility and low bioavailability, while carriers have good water solubility, are not easily destroyed by the intestinal environment, have no toxic side effects and find it easy to form covalent bonds with drugs. 

EGCG is one the most bioactive tea-derived polyphenol constituents [5]. It has attracted immense attention owing to its anti-lipogenic and anti-oxidation effects [6,7]. The distinctive features of EGCG, such as the excellent biopharmaceutical properties and poor bioavailability, affect its efficacy [8]. Researchers have developed various encapsulation techniques to improve the bioavailability of EGCG, including various EGCG-encapsulated nanoparticles (NPs), such as epigallocatechin-3-gallate chitosan nanoparticles (CS-EGCG-NPs) [9], β-lactoglobulin-epigallocatechin-3-gallate nanoparticles (β-lg-EGCG-NPs) [10] and β-lactoglobulin-gum arabic-epigallocatechin-3-gallate nanoparticles (β-lg-GA-EGCG-NPs) [11]. These compounds can potentially help to improve the stability and bioavailability of EGCG [12,13]. However, researchers have primarily focused on the preparation method of nanoparticles and the effect of the cell experiment in vitro. There were few animal studies of nanoparticles in vivo, and the effect of intragastric administration of nanoparticles on the type and abundance of intestinal flora in the prevention model of hyperlipidemic rats has not been studied.

The gut microbiota are known as the most important symbiotic ecosystem inside the body [14,15]. Many metabolic disorders (such as obesity, cardiovascular disease and non-alcoholic fatty liver disease) are significantly influenced by the microbiome present in the human body [16]. It suggested that the intestinal microbiota were the potential therapeutic targets for the treatment of metabolic diseases [17,18]. Therefore, to study the changes in intestinal microflora abundance and species after oral gavage of EGCG nanoparticles, is to provide a theoretical basis for the application of EGCG nanoparticles.

In this work, we prepared EGCG@β-CD NPs by encapsulating EGCG through a self-assembly method using the widely used β-CD as a drug delivery system. We hypothesized that EGCG@ β-CD NPs had a better preventive effect on hyperlipidemia than EGCG via regulating the intestinal microbiome. Metagenomic analysis of gut microbiota composition and quantity was conducted to understand the different action of EGCG@β-CD NPs and EGCG. This study provides new insights into the in vivo effects of nanoparticles.

## 2. Materials and Methods

### 2.1. Preparation of EGCG@β-CD NPs

EGCG@β-CD NPs were prepared by self-assembly processes [19,20]. EGCG was purchased from Hunan Sanfu Biotechnology (Changsha, China) and β-CD was purchased from Sigma Aldrich (St. Louis, MO, USA). Briefly, 20.00 g β-CD (HPLC ≥ 98%) was added to a dry round-bottom flask, followed by 0.01 mol/L pH 7.4 PBS buffer 200 mL. Subsequently, the round-bottom flask was sealed with plastic wrap and fastened using rubber bands. The flask was placed on a shaker (SKY-200B, Shanghai Sukun Industry & Commerce Co. Ltd., Shanghai, China). The contents were heated to 40 °C while being continuously shaken (120 rpm/min; 24 h).

EGCG (20.00 g; HPLC ≥ 98%) and 200 mL distilled water were added to a beaker. The solution was stirred for 1 h using a magnetic agitator to completely dissolve the reagents. According to the 1:1 complex of EGCG and β-CD [21,22], subsequently, it was injected into the β-CD solution using a syringe. The mouth of the container was sealed with a plastic wrap and fastened using rubber bands. A round-bottom flask containing the solution was placed on a shaker and the contents were heated to 40 °C. The contents were continuously shaken at 120 rpm/min for 24 h.

The solution was poured into a 50 mL centrifuge tube and ultrasonicated at 800 W for 5 min using an ultrasonic cell crushing apparatus (VCX800, Sonics). Next, the sample was freeze dried (Alphal-4L SC plus, Martin Christ) for 48 h. Then, the sample was ground through a 120-mesh sieve to obtain powdered EGCG@β-CD NPs. The blank β-CDs were prepared by the same method.

### 2.2. Characterisation of EGCG@β-CD NPs on Embedding Effect

#### 2.2.1. Encapsulation Efficiency (EE)

A 2 mL solution of 1 mg/mL EGCG@β-CD NPs was centrifuged at 4000× *g* for 30 min at 4 °C to remove the supernatant from EGCG@β-CD NPs (pellet) [23]. In addition, the EE was determined by HPLC [24] with some modifications. A chromatographic column (Innoval C18, 5 μm, 4.6 mm × 250 mm, SPD-M20A, SHIMADZU JAPAN) was used. The mobile phases were 15% methanol (A) and 85% water (B, 2% of acetic acid), the flow rate was 1.0 mL/min, column temperature was set at 35 °C and the detection wavelength was 276 nm. EE was calculated by following formula (1).
(1)$$\mathrm{EE}/\%=\frac{\mathrm{EGCG}\;\mathrm{in}\;\mathrm{the}\;\mathrm{pellet},\;\mathrm{mg}}{\mathrm{EGCG}\;\mathrm{in}\;\mathrm{the}\;\mathrm{pellet}+\mathrm{EGCG}\;\mathrm{in}\;\mathrm{the}\;\mathrm{supernatant},\;\mathrm{mg}}\times 100$$

#### 2.2.2. The Particle Size, Polydispersity Index (PDI) and Zeta Potential

Dispersions of the freeze-dried samples of EGCG@β-CD NPs were diluted 200-fold with deionised water and sonicated for 10 min. The mean particle size and PDI were determined using the Particle Size Analyzer (NanoBrook Ommi, Brookhaven Instruments, America) and the zeta potential was determined using Zetasizer Nano ZS90 (Malvern, England).

#### 2.2.3. Scanning Electron Microscopy (SEM)

β-CD and EGCG@β-CD NPs were spread on a double-sided adhesive tape. The excess powder was removed and the sample was sprayed with gold. The morphological features were observed using SEM (QUANTA FEG 250 SUPRA 55 Scanning Electron Microscope, ZEISS, Jena, Germany).

#### 2.2.4. Fourier-Transform Infrared Spectroscopy (FT-IR)

The spectra of the four freeze-dried samples (EGCG, β-CD, physical mixture of EGCG and β-CD, EGCG@β-CD NPs) were recorded using FT-IR (Nicolet iN 10 MX, Thermo Fisher Scientific, USA). The spectral profiles for EGCG, β-CD, physical mixture of EGCG-β-CD and EGCG@β-CD NPs were also recorded. Transparent potassium bromide (KBr) pellets containing the sample were prepared by applying suitable pressure before recording the spectra. The spectra were recorded in a wavelength range of 4000–500 cm^−1^.

#### 2.2.5. X-ray Diffraction (XRD)

The XRD technique was used to analyze the powdered EGCG and β-CD samples. A physical mixture of EGCG-β-CD and EGCG@β-CD NPs was also analyzed. A diffractometer (D8 Advance X-ray Diffraction, Bruker, Germany) was used to record the patterns in a range of 5° to 90° (2θ). The step size was 0.02° and step time was maintained at 5 s. The XRD curve of each sample was recorded at voltage and current 40 kV and 30 mA, respectively. The target/filter (monochromator) was copper.

#### 2.2.6. Molecular Docking

The tridimensional structure of EGCG (PubChem CID: 65064) was obtained from the PubChem database (https://pubchem.ncbi.nlm.nih.gov/, accessed on 25 May 2022). The crystal structures of β-CD (PDB ID: 3cgt) were downloaded from the Research Collaboratory for Structural Bioinformatics Protein Data Bank (http://www.rcsb.org/pdb/home/home.do, accessed on 2 August 2020). The data corresponding to the optimised molecules were saved as pdf files. The data were input using AutoDockTool 1.5.6 (http://autodock.scripps.edu/, accessed on 3 August 2020). All atom types were assigned, and all the hydrogen atoms were added to AutoDock. The rotatable bonds were set as “flexible” and the data were then converted to PDBQT format to conduct the docking studies. Docking studies were performed using AutoDock vina 1.1.2 (http://autodock.scripps.edu/, accessed on 4 August 2020). The docking results were analyzed and the best affinity from the conformation was chosen to conduct the subsequent molecular dynamics studies.

### 2.3. Animals Assay 

#### 2.3.1. Experimental Design

All animal experiments should comply with the ARRIVE guidelines and were carried out in accordance with the U.K. Animals (Scientific Procedures) Act, 1986, and associated guidelines, EU Directive 2010/63/EU for animal experiments. Healthy Specific Pathogen-Free grade four-week-old male Sprague–Dawley (SD) rats weighing in a range of 150 g to 180 g were supplied by the Hunan SJA Laboratory Animal Co., Ltd. (Changsha, China). The permit animal certification number was SCXK (Xiang) 2019-0004 and the animal registry number was 1107272011005688. The assays were conducted at the Tea Research Institute of Hunan Agricultural University (Changsha, China). These rats were fed in a barrier environment. The humidity was maintained in a range of 50–60% and the temperature was maintained in a range of 23–25 °C. The experiments were conducted under conditions of lighting (14 h light/10 h darkness). Four rats were housed in one cage. Food and water were supplied ad libitum. The weight of the leftover feed was recorded each day and the rats were weighed once a week for calculating food utilization using formula (2). All animal-based experiments were approved by the local authorities. The experiments were conducted following the regulations laid down by the Care and Use of Laboratory Animals.
(2)$$\mathrm{Food}\;\mathrm{utilisation}\;(\%)=\frac{daily\;body\;weight\;gain}{daily\;food\;intake}\times 100$$

After one week of adaptive feeding, 48 rats were randomly divided into 6 groups according to the diet, with 8 rats in each group. Each group was labelled: (a) ND group: normal diet (10 kcal % fat, TP23302, TROPHIC Animal Feed High-tech Co., Ltd., Nantong, China); (b) HFD group: high-fat diet (60 kcal % fat, TP23300, TROPHIC Animal Feed High-tech Co., Ltd., Nantong, China); (c) SIM group: high-fat diet + 5 mg/kg·bw simvastatin (purity 99.83% Abmole Bioscience Inc., Houston, TX, USA); (d) E group: high-fat diet +100 mg/kg·bw EGCG; (e) BB group: high-fat diet +100 mg/kg·bw β-CD; and (f) BE group: high-fat diet +200 mg/kg·bw EGCG@β-CD NPs (100 mg/kg·bw EGCG). All the reagents were dissolved in a saline solution to prepare 1 mL of the gavage. 

#### 2.3.2. Sample Collection and Preparation

Feces were collected from 9:00–11:00 three days before the end of the experiment. Feces of each rat were collected in a sterile freezing tube, immediately placed in liquid nitrogen and stored in a refrigerator at −80 °C for metagenomic analysis. After 8 weeks of treatment, rats were slaughtered. Blood samples were collected through heart into 15 mL centrifuge tubes, placed at room temperature for 1 h, centrifuged (3000× *g*, 15  min, 4 °C) and then detected the serum cholesterol (TC), triglycerides (TG), high-density lipoprotein cholesterol (HDL-C) and low-density lipoprotein cholesterol (LDL-C) indexes. Hepatic and adipose tissue was dissected and weighed accurately. Some samples were fixed in 4% paraformaldehyde. The remaining hepatic and adipose tissue was immediately placed in liquid nitrogen and stored in a refrigerator at −80 °C for subsequent studies.

#### 2.3.3. Biochemical Analysis and Histological Examination

The levels of TC, TG, HDL-C, LDL-C, phospholipids (PL) in serum and the levels of TC, TG, liver malondialdehyde (MDA), superoxide dismutase (SOD), catalase (CAT), glutathione peroxidase (GSH-Px) and apolipoprotein-A1 (apo-A1) in liver were determined using a biochemical kit assay (Nanjing Jiancheng Bioengineering Institute, Nanjing, China) by reference to the manufacturer’s instructions. 

Hepatic and adipose tissue was fixed with 4% paraformaldehyde for 48 h at 4 °C, embedded in paraffin, prepared sections (4 μm thick), counterstained with hematoxylin and eosin (H&E) for histological analysis. Following this, the images were recorded using 3DHISTECH (3D HISTECH Ltd., Budapest, Hungary) and analyzed using the Case Viewer software.

### 2.4. Metagenomics Analysis

#### 2.4.1. Library Construction and Sequencing

Desoxyribonucleic acid (DNA) degradation degree and potential contamination were monitored on 1% agarose gels. DNA concentration was measured using Qubit^®^ dsDNA Assay Kit in Qubit^®^ 2.0 Flurometer (Life Technologies, CA, USA). The Optical density (OD) value was approximately between 1.8 and 2.0, DNA contents above 1µg were used to construct library. Sequencing libraries were generated using NEBNext^®^ Ultra™ DNA Library Prep Kit for Illumina (NEB, USA) and index codes were added to attribute sequences to each sample. The clustering of the index-coded samples was performed on a cBot Cluster Generation System. After cluster generation, the library preparations were sequenced on an Illumina HiSeq platform and paired-end reads were generated. The raw data were obtained using Readfq (V8, https://github.com/cjfields/readfq, accessed on 18 October 2020) and processed to clean data for subsequent analysis. 

#### 2.4.2. Metagenome Assembly

The clean data were assembled and analyzed [25] by Short Oligonucleotide Analysis Package (SOAP) denovo software (V2.04, http://soap.genomics.org.cn/soapdenovo.html, accessed on 11 November 2020), then interrupted the assembled Scaftigs from N connection and left the Scaftigs without N [26]. All samples’ clean data were compared to each Scaffold, respectively, by Bowtie2.2.4 software to acquire the PE reads not used. All the reads not used in the forward step of all samples were combined and then used the software of SOAP denovo (V2.04)/MEGAHIT (v1.0.4-beta) for mixed assembly with the parameters, same as for single assembly. Break the mixed assembled Scaffolds from N connection and obtained the Scaftigs. Filter the fragment shorter than 500 bp in all of Scaftigs for statistical analysis both generated from single or mixed assembly.

#### 2.4.3. Gene Prediction and Abundance Analysis

The open reading frame (ORF) predicted the Scaftigs (≥500 bp) assembled from both single and mixed by MetaGeneMark (V2.10, http://topaz.gatech.edu/GeneMark/, accessed on 23 December 2020) software, and filtered the length information shorter than 100 nt [27,28] from the predicted result with default parameters. For predicted ORF, CD-HIT [29] software (V4.5.8, http://www.bioinformatics.org/cd-hit, accessed on 24 December 2020) was adopted to redundancy and obtained the unique initial gene catalogue [30]. The clean data of each sample were mapped to initial gene catalogue using Bowtie2.2.4, which obtained the number of reads to which genes were mapped in each sample [28,31]. We then filtered the genes when the number of reads ≤ 2 [32] in each sample and obtained the gene catalogue (genes) eventually used for subsequent core-pan gene analysis, correlation analysis of samples and Venn figure analysis. 

#### 2.4.4. Taxonomy Prediction

DIAMOND software (V0.9.9, https://github.com/bbuchfink/diamond/, accessed on 9 June 2021) was used to blast the genes to the sequences of Bacteria, Fungi, Archaea and Viruses which were all extracted from the non-redundant (NR) database (Version: 2018-01-02, https://www.ncbi.nlm.nih.gov/, accessed on 11 June 2021) of NCBI. Choosing the result of which the e value ≤ the smallest e value *10 [33] to take the lowest common ancestry (LCA) algorithm which was applied to system classification (kingdom, phylum, class, order, family, genus, species) of MEGAN software ensured the species annotation information of sequences. Krona analysis, the exhibition of generation situation of relative abundance and principal component analysis (PCA) (R ade4 package, Version 2.15.3) decrease-dimension analysis were based on the abundance table of each taxonomic hierarchy. Metastats and linear discriminant analysis (LDA) effect size (LEfSe) analysis were used to look for the different species between groups. Permutation test between groups was used in Metastats analysis for each taxonomy and obtained the P value, then used Benjamini and Hochberg False Discovery Rate to correct P value and acquire q value. LEfSe analysis was conducted by LEfSe software (the default LDA score was 4) [34]. 

### 2.5. Statistical Analysis

Statistical significance was analyzed by conducting one-way analysis of variance (ANOVA) tests using GraphPad Prism software 7.0 (La Jolla, CA, USA), followed by Tukey’s multiple-comparison test. With a normalized relative abundance matrix, LEfSe used the Kruskal–Wallis rank sum test to detect features with significantly different abundances between assigned taxa and performed LDA to estimate the effect size of each feature. A significance alpha of 0.05 and an effect size threshold of 3 were used for the biomarkers discussed in this study.

## 3. Results

### 3.1. Characterisation of the EGCG@β-CD NPs on Size, PDI, EE and Zeta Potential

Normal distribution was observed for the diameter of the EGCG@β-CD NPs. Further, the average particle diameter was calculated to be 124.6 nm (Figure 1A). The PDI values were in a range from 0 to 1, with stable disperse system closed to zero [35]. The particle size distribution was divided into small span values (PDI = 0.313). The results indicated that the EGCG@β-CD NPs exhibited in homogeneity and a uniform particle size distribution. Furthermore, the EE measured by HPLC was 98.27 ± 0.36%.

The higher the absolute values of the zeta potential, the higher repulsion faced by the NPs, and more stable the dispersed system [36]. The zeta potential of the EGCG@β-CD NPs was –24.3 mV (Figure 1B). It is generally considered that the system is stable when the absolute value of Zeta potential is greater than 30mV, but the particles embedded with β-CD often have steric hindrance, which makes the system stable even at a lower Zeta potential. The results revealed that the dispersed systems were good under these conditions. 

### 3.2. Characterisation of the EGCG@β-CD NPs on the Particle Morphology

As shown in Figure 1C, SEM images of EGCG@β-CD NPs and β-CD (×10,000, upper-i and ii; and ×50,000, lower-iii and iv) were recorded. The micrographs showed that the β-CD was irregular as plates and strips. However, the shapes of EGCG@β-CD NPs were observed to be more uniform and granular.

### 3.3. Characterisation of the EGCG@β-CD NPs on FT-IR

The EGCG molecule bears three benzene rings, a non-aromatic six-membered ring and an ester group (Figure 1D). The absorption peak at 3477.66 cm^−1^ and 3356.14 cm^−^^1^ could be attributed to the stretching vibrations of the phenolic hydroxyl groups present in the benzene ring. The peak corresponding to CO_2_ appeared at 2338.68 cm^−1^. The peak at 1691.57 cm^−^^1^ was attributed to the C=O stretching vibration of the ester bonds. Multiple absorption peaks appeared to be in a range of 1000–1600 cm^−1^. The peaks were ascribed to various vibrations observed in the benzene ring. The peaks at 1346.31 cm^−^^1^ and 1292.31 cm^−1^ could be ascribed to the stretching vibrations of the methylene group. The characteristic peaks of β-CD appeared at 3383.14 cm^−^^1^, 2911.26 cm^−1^, 1641.42 cm^−1^ and 1028.06 cm^−1^, respectively. The peaks correspond to the symmetric and antisymmetric bending vibrations of the OH group, stretching vibrations of the CH_2_ group, stretching vibrations of the C=O group and bending vibrations of the C–O group, respectively (Figure 1D-ii). Characteristic peaks of EGCG and β-CD were also observed in the spectral profiles (Figure 1D-iii). The peaks corresponding to EGCG appeared at 3475.73 cm^−1^, 3351.14 cm^−1^ and 1691.57 cm^−1^. The peaks at 2926.01 cm^−1^, 1620.21 cm^−1^ and 1031.92 cm^−1^ could be assigned to β-CD. Analysis of the spectra profiles revealed that the samples were a physical mixture of EGCG and β-CD. The number of peaks in the curve presented in Figure 1D-iv was lower than the number of peaks presented in Figure 1D-iii. The intensities of the peaks in the profile presented in Figure 1D-iv were lower than the strength of the peaks presented in the profiles presented in Figure 1D-iii. The peaks presented in Figure 1D-iv appeared broader than those presented in Figure 1D-iii. The peaks corresponding to the C-H stretching of methylene groups in the EGCG-recorded profiles disappeared in the EGCG@β-CD NP-recorded profiles. This can be attributed to the fact that the presence of a phenyl ring in EGCG entrapped into the β-CD cavity during the formation of the inclusion complex. The appearance of new peaks was not observed in the inclusion complex, indicating independent status between EGCG and β-CD. In the FT-IR analysis, the varying of the peak position, the evaporation of some peaks and the declination of peak intensity indicated that EGCG was encapsulated in the β-CD hydrophobic nanocavity.

### 3.4. Characterisation of the EGCG@β-CD NPs on XRD

The structures of EGCG, β-CD, EGCG@β-CD NPs and the physical mixture consisting of EGCG and β-CD (Figure 1E) were characterized by XRD. EGCG appeared at Bragg angles (2θ) of 15.56°, 16.98°, 20.66°, 21.48°, 24.48° and 25.86° (Figure 1E-i). The physical mixtures of EGCG and β-CD were recorded (Figure 1E-iii). The peaks that appeared at 15.56°, 20.66° and 21.48° corresponded to EGCG (Figure 1E-i) and the peaks that appeared at 12.4°, 31.64° and 45.38° to β-CD (Figure 1E-ii). These peaks were absent or low-intensity peaks were presented in these regions in the profiles recorded for EGCG@β-CD NPs (Figure 1E-iv). A lower number of peaks in the profile recorded for the EGCG@β-CD NPs indicated an association between EGCG and β-CD. The results confirmed that the β-CD NPs could be effectively encapsulated with EGCG.

### 3.5. Characterisation of the EGCG@β-CD NPs on Molecular Docking

Results from the docking experiments revealed that EGCG (Figure 1F-i) could interact with β-CD (Figure 1F-ii). The docking score was calculated to be –6.8 kcal/mol. EGCG and β-CD interacted with each other through stable *π*–*π* stacking interactions (Figure 1F-iii) The benzene rings at both ends of EGCG were observed to open toward the large cavity, while the middle benzene ring opened toward the smaller cavity. Results from the structural analysis revealed that EGCG was entrapped within β-CD (Figure 1F-iv). Eight hydrogen bonds helped in encapsulation (Figure 1F-iii). The three rings in EGCG were involved in the formation of the β-CD inclusion complex due to the less steric hindrance they faced (Figure 1F-i,iv) [37].

### 3.6. Effect of EGCG@β-CD NPs on Body Weight, Food Intake, Weight Gain, Adiposity Index and Food Utilization in Rats Fed with High-Fat Diet

To further validate the effect of EGCG@β-CD NPs in vivo, high-fat-diet-fed rats were treated with various intervention programs for eight weeks. As shown in Figure 2A, during the 8-week intervention feeding, body weights increased to varying degrees in each group. In the high-fat-diet-fed group, the weight gain in the BE group was significantly different from that in the HFD group, E group and SIM group (*p* < 0.05), and approached the ND group (Figure 2C).

As shown in Figure 2B, food intake in each group showed a decreasing trend over 1–3 weeks, possibly due to discomfort caused by oral gavage treatment. With the adaptation of the gavage and the needs of growth, the food intake began to increase continuously after 3–8 weeks. Although food intake in the E and BE groups decreased after 6 weeks, there were no significant difference in food intake in the high-fat-diet rats over 8 weeks (Figure 2D). The result is in line with [38].

As shown in Figure 2F, compared with the ND group, the food utilization rate in the other high-fat-diet groups was significantly increased (*p* < 0.05), except for the BE group. Compared with the other high-fat-diet groups, the BE group had lower food utilization. Meanwhile, the adiposity index (Figure 2E) also showed the same results as food utilization. This indicated that EGCG@β-CD NPs reduced food utilization, slowed weight gain and reduced adipose formation in high-fat-diet rats without affecting food intake.

### 3.7. Effect of EGCG@β-CD NPs on Liver Injury in Rats Fed with High-Fat Diet

As shown in Figure 3A, the H&E staining of rat liver tissue revealed that substantial lipid droplet vacuoles were observed in the liver in the HFD group and the liver represented obvious degeneration. Additionally, the white adipose in the HFD group markedly enlarged the size (Figure 3B). However, these changes were suppressed to varying degrees in all intervention groups, with the supplementation of EGCG@β-CD NPs (BE group) closest to the ND group. This indicated that gavage of EGCG@β-CD NPs showed a better protective effect on the liver compared with EGCG.

### 3.8. Effect of EGCG@β-CD NPs on Lipid Accumulation and Peroxidation in Rats Fed with High-Fat Diet

As shown in Table 1, compared with the ND group, the serum TC, LDL-C and PL were significantly higher in the HFD group (*p* < 0.05), whereas the intervention of EGCG@β-CD NPs caused marked reductions in these indices. Nevertheless, the apo-A1 showed an opposite trend (*p* < 0.05). In contrast, the EGCG@β-CD NPs did not reveal an obvious effect on the HLD-C after the administration of EGCG@β-CD NPs for 8 weeks. It was noteworthy that EGCG@β-CD NPs’ intervention also dramatically retarded the increase in TC levels in the liver, caused by the high-fat diet (*p* < 0.05).

Meanwhile, the high-fat diet markedly enhanced the liver MDA contents, but EGCG@β-CD NPs inhibited this elevation (*p* < 0.05). Supplementation with EGCG@β-CD NPs significantly reversed the decrease in hepatic CAT activities caused by lipid peroxidation. Moreover, compared with the ND group, SOD was significantly different in the liver in the high-fat-diet group (*p* < 0.05), while the difference was not significant among the high-fat-diet groups. GSH-Px levels in the HFD group were significantly different from those in the ND group (*p* < 0.05), but not significantly different from other high-fat-diet groups.

### 3.9. Influence of EGCG@β-CD NPs on Gut Microbiome Diversity in Rats Fed with High-Fat Diet

As shown in Figure 4A, with the increasing number of sequencing samples, the curve first rose and then flattened, indicating that the species in the environment increased with the increase in the number of samples, but when the number of samples increased to a certain extent, the number of species did not increase significantly and tended to be saturated. The sample size was sufficient to image the richness of the community. In addition, the analysis revealed that only 270,008 of the total richness of 785,757 operational taxonomic units (OTUs) were shared among all samples (Figure 4B). The data suggested that over 34% of the observed OTUs after EGCG@β-CD NPs feeding was the same as in the initial treatment.

### 3.10. Effect of EGCG@β-CD NPs on the Bacterial Variety in Rats Fed with High-Fat Diet

At the phylum level (Figure 5A), the most abundant intestinal microbiome community included *Firmicutes, Bacteroidetes, Verrucomicrobia* and *Proteobacteria*. After 8 weeks of EGCG@β-CD NPs intervention, the relative abundance of *Verrucomicrobia* and *Tenericutes* heightened significantly from 4.85 ± 0.01 to 8.58 ± 0.02, 0.38 ± 0.01 to 0.85 ± 0.01, respectively, while *Bacteroidetes* revealed the opposite tendency. 

The top 10 of the relative abundance of intestinal flora in rats, at the order level, are shown in Figure 5B. It can be known that EGCG@β-CD NPs had active regulating affection on *Clostridiales* and *Bacteroidales*. After 8 weeks of EGCG@β-CD NP administration, the relative abundance of *Verrucomicrobia*, *Lactobacillales* and *Eggerthellales* increased, whereas, the relative abundance of *Bacteroidales* and *Enterobacterales* decreased remarkably. Moreover, *Caudovirales* and *Mycoplasmatales* also attracted attention. These were in accordance with the shift in intestinal flora abundance at the phylum level.

The primary bacterial groups of rats in each group at the family level were *Lachnospiraceae*, *Ruminococcaceae*, *Bacteroidaceae* and *Akkermansiaceae* (Figure 5C). *Lachnospiraceae* was the most predominant type, with a relative abundance of 17.21%, and the other strains were less than 10%. Compared with the HFD group, the EGCG@β-CD NP group showed an increased abundance of *Ruminococcaceae* and *Akkermansiaceae*. Nevertheless, *Bacteroidaceae*, *Tannerellaceae, Eubacteriaceae* and *Prevotellaceae* exhibited exponential declines. There was little difference between each group in the abundance of *Lachnospiraceae*, *Clostridiaceae*, *Oscillospiraceae* and *Desulfovibrionaceae.*

At the species level, according to Figure 5D, the sum of the top 10 OTUs was no more than 15%. The EGCG@β-CD NP group was characterized by a higher abundance of *Akkermansia muciniphila* and a lower abundance of *Lachnospiraceae bacterium 28-4*, *Bacteroides sartorii*, *Oscillibacter sp. 1-3* in comparison to the HFD group. Noteworthily, *Lachnospiraceae bacterium 10-1* in EGCG group was significantly higher than that in other groups at the species level.

To study the species with significant differences between groups, PCA revealed a clear interval between the HFD group and high-fat-diet-treated groups, evidencing that EGCG and EGCG@β-CD NPs showed a significant impact on the gut microbiota of the hyperlipidemia rat at the genus level. As shown in Figure 6, in the top 12 genera, a higher abundance of *Gracilimonas* (A), *Coprobacter* (B), *Carnobacterium* (D), *Butyricimonas* (F), *Alloprevotella* (G), *Agrobacterium* (H), *Rhodobacter* (I), *Millionella* (J) and *Ruminococcus* (K) in the ND group were demonstrated as compared to the BE group (q < 0.05). Meanwhile, EGCG treatment induced the increasing abundances of *Fretibacterium* (L) (q < 0.05) and the decreasing abundances of *Cupriavidus* (C) (q < 0.01) and *Alloprevotella* (G) (q < 0.05). 

### 3.11. LEfSe Analysis

To recognize the specific bacterial taxa related to the intervention of EGCG@β-CD NPs, the gut microbiota in the high-fat diet-induced hyperlipidemia rat model were exhibited by LEfSe. Taxa with an LDA score threshold > 4 were contained in the LEfSe and counts of very-low-abundance bacteria were ignored (Figure 7A). Studies found that the ND group was dominated by the order *Bacteroidales*. Different treatments of high-fat diets had distinct marker bacteria. The HFD group was largely characterized by the species *Blautia_sp_CAG_257*, *Bacteroides_caecimuris*, *Anaerotruncus_colihominis*, *Bacteroides_congonensis* and *Bacteroides_fragilis*. The E group was characterized by the species *Clostridium_sp_CAG_169*, *Lachnoclostridium_Clostridium_bolteae, Adlercreutzia_equolifaciens*, *Flavonifractor_sp_An306*, *Firmicutes_bacterium_CAG_555, Oscillibacter_sp_PC13*, *Clostridiales_bacterium_1_7_47FAA*, *Oscillibacter_sp_CAG_155*, *Firmicutes_bacterium_CAG_103* and *Clostridiales_bacterium_VE202_28*, whereas the species *Gemmiger_sp_An120*, *Streptococcus_gallolyticus*, *Fournierella_massiliensis*, *Lachnoclostridium_sp_An196*, *Phascolarctobacterium_sp_CAG_207*, *Firmicutes_bacterium_CAG_145*, *Gemmiger_sp_An194*, *An87*, *An50*, *Anaerofilum_sp_An201*, *Erysipelatoclostridium_Clostridium_saccharogumia*, *Subdoligranulum_variabile*, *Collinsella_stercoris* and *intestinalis* were the highest proportion bacteria in the BE group. This indicated that the microbial species differed greatly between the different treatments at the species level.

As shown in Figure 7B, at the phylum level, *Bacteroidetes* revealed marked differences in the gut microbiota from HFD group, while Unclassified Bacteria were significantly enriched in the ND group. No differences were shown for biomarkers in treatment groups when fed with high-fat diet for 8 weeks.

## 4. Discussion

EGCG@β-CD NPs were polymers through host–guest interactions. This complex greatly enhanced the stability of EGCG due to the multi-point interactions between EGCG and β-CD [39]. Molecular docking provided a detailed molecular representation of NPs based on the binding process of EGCG-β-CD. EGCG was considered as a targeted component for the inclusion of β-CD, which improved the efficacy of EGCG delivery [40]. This may be due to the polymer form of EGCG@β-CD NPs, which should be hydrolyzed by intestinal enzymes or colonic microbiota before they can be absorbed [41]. In general, the mechanisms by which encapsulated bioactives can be released in the large intestine are (i) due to the presence of a large amount of fluid in the stomach and small intestine, the encapsulated bioactives have low mechanical resistance, and after reaching the colon, they have strong peristaltic waves, and the increased intraluminal pressure leads to rupture; (ii) the carrier is a pH-sensitive polymer that remains intact in the stomach but is easily digested by enzymes in specific parts of the gut; (iii) changes in the microbiota cause changes in enzyme activity that target the gut to release biologically active substances [42]. Continuous studies have confirmed that polyphenols have a positive impact on intestinal health by biotransforming them into bioactive metabolites and/or modulating gut microbiota composition [43]. Collectively, these polyphenol metabolites may be essential for sustaining normal intestinal microbiota composition and contribute significantly to improving host health [44].

The in vivo effects of EGCG@β-CD NPs following oral administration to rats with diet-induced hyperlipidemia were investigated and compared with those of EGCG.

The results showed that the effect of EGCG@β-CD NPs on reducing the adiposity index in hyperlipidemic rats was better than free EGCG. Furthermore, our results exhibited that the treatment of EGCG@β-CD NPs could significantly decrease TC, TG and MDA and significantly increase CAT and apo-A1 in liver. In addition, the accumulation of free radicals can lead to oxidative stress and cause the occurrence of chronic metabolic diseases [45]. Lipid oxidation levels were elevated in high-fat-diet rats. A better anti-hyperlipidemic effect was revealed for EGCG@β-CD NPs in comparison with EGCG by reducing the contents of TC, TG and MDA and increasing the levels of SOD, CAT, GSH-Px and apo-A1 in the liver. In addition, the H&E staining of liver sections showed that EGCG@β-CD NPs could better reduce the accumulation of lipid droplets in the liver. Therefore, these results exhibited that the oral administration of EGCG@β-CD NPs could better interfere with hepatic oxidative damage and fat accumulation using a high-fat diet, reducing the levels of TC, LDL-C and PL in the blood.

Growing evidence indicates the important role of the gut microbiota in lipid metabolism disorders. The gut microbiota contributed to metabolic syndrome via affecting the host’s energy metabolism, immune system and inflammatory response [46]. Amanda J Cox et al. indicated that a disturbance of gut microbiota and a change in intestinal permeability were potential triggers of inflammation in obesity [47]. The homeostasis of gut microbiota is regulated by both its internal genes and external factors, such as diet, alcohol, exercise and drugs, among which diet has the greatest impact [48]. We found that feeding high-fat diets for 8 weeks evidently disrupted the balance of the gut microbiota, greatly lowered the diversity of intestinal microflora and caused dyslipidemia and metabolic disorder, which were consistent with previous studies [49]. Therefore, dietary intervention was considered to be one of the important ways to prevent and treat dyslipidemia.

Furthermore, the results revealed that EGCG@β-CD NP treatment increased the abundance of phylum *Verrucomicrobia*, especially *Akkermansia muciniphila*, indicating that EGCG may alter the abundance of *Verrucomicrobia* in high-fat-diet-fed rats. In addition, EGCG significantly increased the relative abundance of *Akkermansia muciniphila*, which prevented the deleterious effects of a high-fat diet in mice [50,51]. This may also be the reason why the liver slice damage was minimal in the BE group after gavage of EGCG@β-CD NPs. Of course, the strain changes induced by tea polyphenols varied with dose, treatment time and animals [52]. According to Liu et al., the relative abundance of *Akkermansia muciniphila* was significantly increased after 12 weeks of administration of 210 mg/kg EGCG to high-fat-diet-fed C57BL/6J mice [53]. However, this paper found that the enrichment effect of *Akkermansia muciniphila* was not obvious when SD rats were given 100 mg/kg EGCG daily for 8 weeks. When 200 mg/kg EGCG@β-CD NPs were intragastrically administered, that is, containing 100 mg/kg EGCG, the relative abundance of *Akkermansia muciniphila* was significantly increased. Meanwhile, the possibility that β-CD increased the relative abundance of *Akkermansia muciniphila* was also excluded. Therefore, EGCG@β-CD NPs may enhance the content of EGCG reaching the intestine, compared with free EGCG.

Moreover, a high-fat diet has been confirmed to alter the intestinal microbiota and cause inflammation [54], while *Akkermansia muciniphila* can prevent obesity induced by a high-fat diet and inhibit the occurrence of complications [55]. This may be related to the fact that the proteins encoded in the genome of *Akkermansia muciniphila* contain signal peptides and these genes may have a strong ability to break down extracellular polymeric substrates, such as mucins [56]. The relative abundance of *Akkermansia muciniphila* was influenced by the processes of energy loss and thermogenesis [57,58,59]. The properties of *Akkermansia muciniphila* do not affect lipid absorption or chylomicron synthesis, but rather increase fecal energy content [55] and decrease food energy efficiency [60]. Consistent with the results, gavage of EGCG@β-CD NPs reduced food utilization, slowed weight gain and reduced adipogenesis in high-fat-diet rats without affecting food intake. Hence, *Akkermansia muciniphila* was often used as a marker for a healthy microbiome [61]. However, the mechanism of *Akkermansia muciniphila’s* regulation by EGCG and its regulation on hyperlipidemic rats need to be further studied.

## 5. Conclusions

In this work, the results showed that EGCG could be encapsulated with β-CD to form EGCG@β-CD NPs, which exhibited superior blood-lipid-lowering and liver protection effects in comparison with direct gavage EGCG. Moreover, the metagenome indicated that gavage of EGCG@β-CD NPs and free EGCG had different effects on the structure and abundance of the gut microbiome. After 8 weeks of EGCG@β-CD NP intervention, the ratio of *Verrucomicrobia/Bacteroidetes* in the intestinal microbiota of rats was changed, *Bacteroidetes* decreased and *Verrucomicrobia* increased significantly, especially *Akkermansia muciniphila*, which was not significant in direct gavage of EGCG. However, in order to understand the different roles of EGCG@β-CD NPs and EGCG, the mechanism of EGCG@β-CD NPs and EGCG regulating the species and number of microorganisms needs to be further studied.

## Figures and Tables

**Figure 1 molecules-27-04894-f001:**
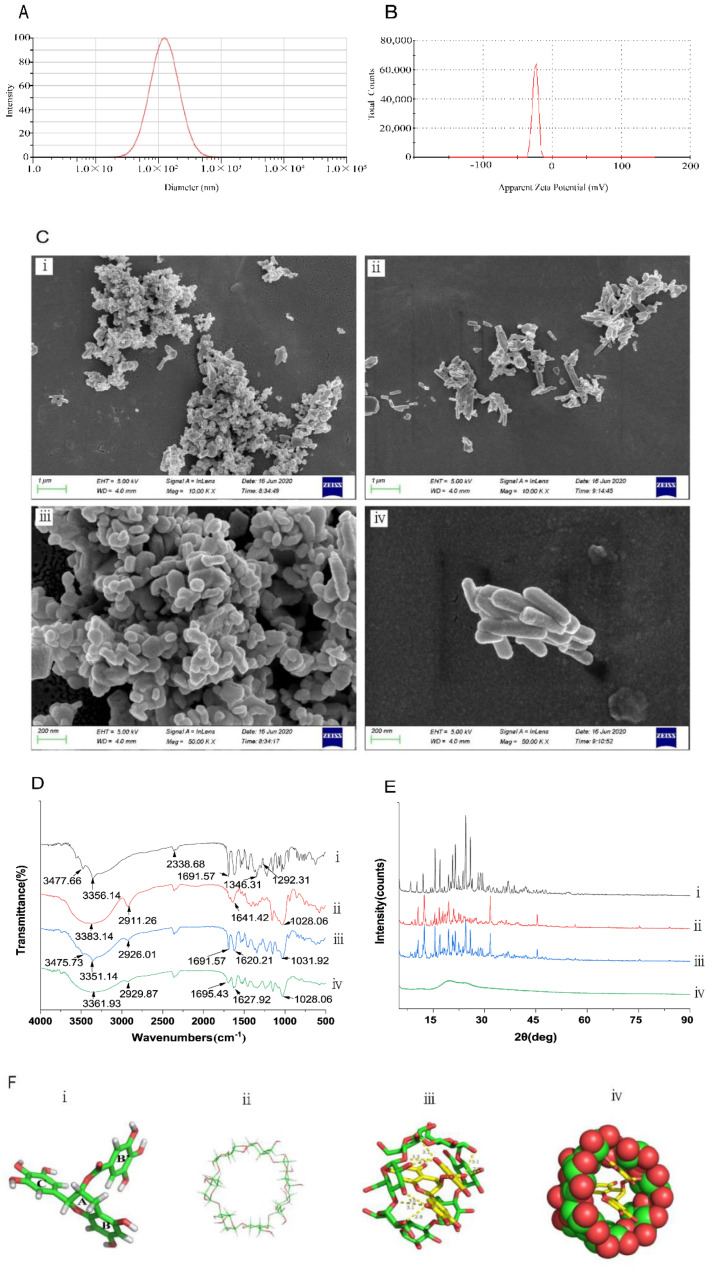
Characterization of EGCG@β-CD NPs. (**A**) Size distribution of EGCG@β-CD NPs. (**B**) zeta potential distribution of EGCG@β-CD NPs. (**C**) SEM images of the EGCG@β-CD NPs and β-CD. (i) and (ii) represent the SEM images (magnification ×10,000) of the EGCG@β-CD NPs and β-CD, respectively. (iii) and (iv) represent the SEM images (magnification ×50,000) of the EGCG@β-CD NPs and β-CD, respectively. (**D**) FT-IR spectra. (i) EGCG, (ii) β-CD, (iii) physical mixture of EGCG and β-CD, and (iv) EGCG@β-CD NPs. (**E**) XRD. (i) EGCG, (ii) β-CD, (iii) physical mixture of EGCG and β-CD, and (iv) EGCG@β-CD NPs. (**F**) Molecular docking. (i) 3D structure of EGCG, (ii) 3D structure of β-CD, (iii) molecular force of EGCG and β-CD, and (iv) 3D structure of EGCG@β-CD NPs.

**Figure 2 molecules-27-04894-f002:**
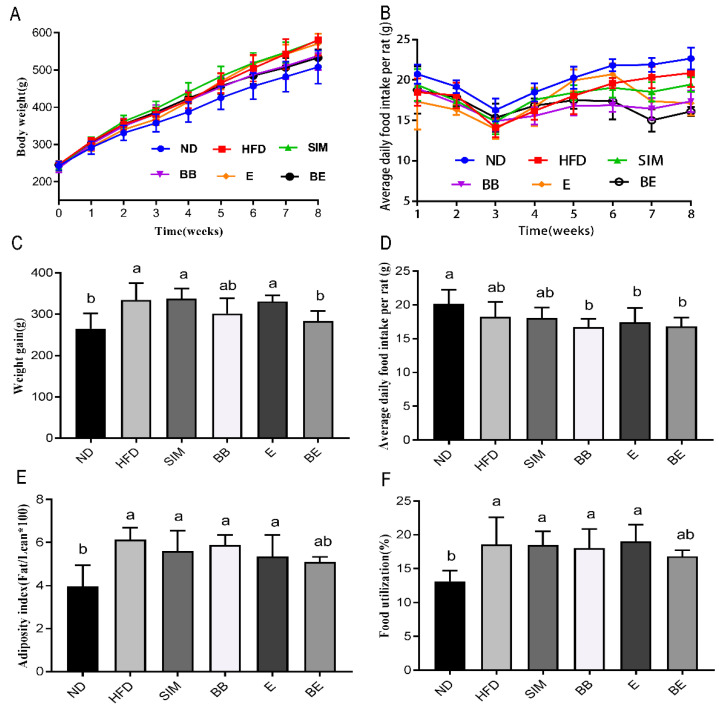
Effects of EGCG@β-CD NPs and EGCG on body weight, food intake, weight gain, adiposity index, and food utilization. (**A**) Body weight. (**B**) Average daily food intake per rat. (**C**) Weight gain. (**D**) Average daily food intake per rat. (**E**) Adiposity index. (**F**) food utilization. All results are expressed as mean ± SE (*n* = 8 rats/group), Superscript characters indicate significant differences between treatments. ND: normal diet + purified water; HFD: high-fat diet + purified water; SIM: high-fat diet + 5 mg/kg·bw simvastatin; E: high-fat diet + 100 mg/kg·bw EGCG; BB: high-fat diet + 100 mg/kg·bw β-CD; BE: high-fat diet + 200 mg/kg·bw EGCG@β-CD NPs (100 mg/kg·bw EGCG).

**Figure 3 molecules-27-04894-f003:**
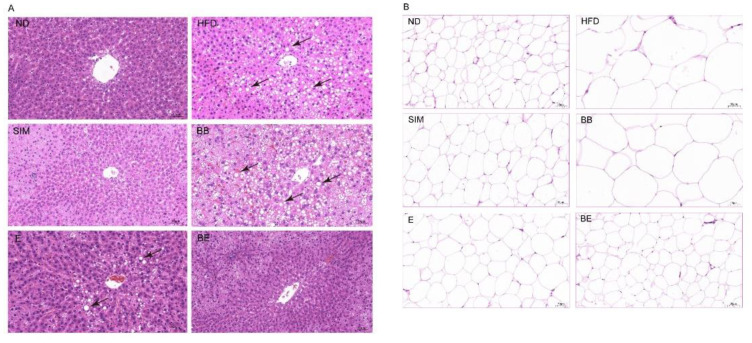
Effects of EGCG@β-CD NPs and EGCG on alleviation of hepatic lipid accumulation in high-fat diet-induced hyperlipidemic rats. (**A**) Liver steatosis ratio stained with hematoxylin and eosin (H&E). (**B**) White adipose stained with H&E. Scale bar = 50 µm. All results are expressed as mean ± SE (*n* = 8 rats/group). ND: normal diet + purified water; HFD: high-fat diet + purified water; SIM: high-fat diet + 5 mg/kg·bw simvastatin; E: high-fat diet +100 mg/kg·bw EGCG; BB: high-fat diet + 100 mg/kg·bw β-CD; BE: high-fat diet + 200 mg/kg·bw EGCG@β-CD NPs (100 mg/kg·bw EGCG).

**Figure 4 molecules-27-04894-f004:**
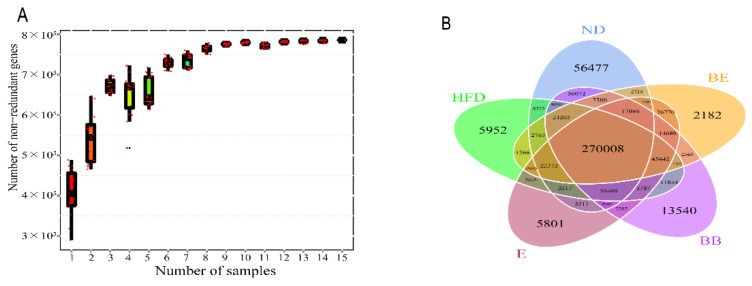
(**A**) Number of bacterial OTUs derived from each sample. (**B**) Venn diagram showing the overlap of OTUs in gut microbiota among the samples. ND: normal diet + purified water; HFD: high-fat diet + purified water; E: high-fat diet +100 mg/kg·bw EGCG; BB: high-fat diet + 100 mg/kg·bw β-CD; BE: high-fat diet + 200 mg/kg·bw EGCG@β-CD NPs (100 mg/kg·bw EGCG).

**Figure 5 molecules-27-04894-f005:**
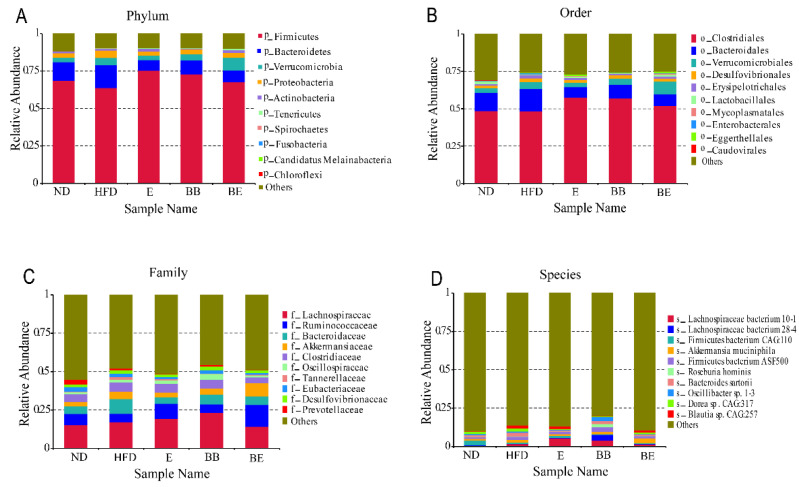
(**A**–**D**) Relative abundance analysis at the phylum, order, family and species level in all groups, with different colored bars standing for different gut microbiota compositions. ND: normal diet + purified water; HFD: high-fat diet + purified water; E: high-fat diet + 100 mg/kg·bw EGCG; BB: high-fat diet + 100 mg/kg·bw β-CD; BE: high-fat diet + 200 mg/kg·bw EGCG@β-CD NPs (100 mg/kg·bw EGCG).

**Figure 6 molecules-27-04894-f006:**
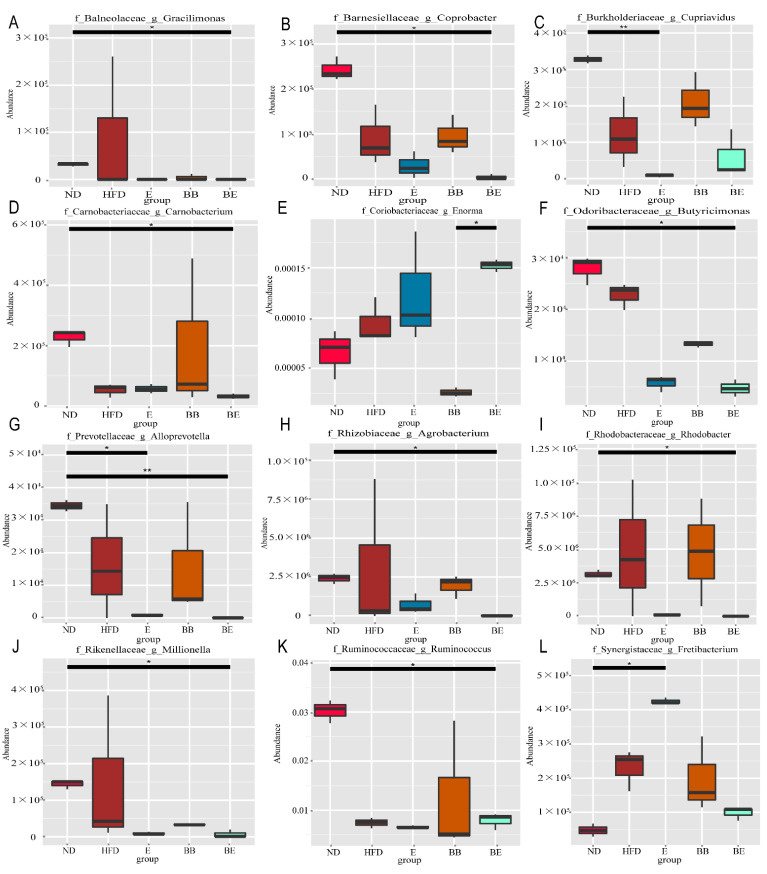
Significantly different taxa displayed in box diagram at genus level. Significantly varying taxa have been identified by t-test/ANOVA (adjusted *p*-value (q) < 0.05). At genus level, the Gracilimonas (**A**), Coprobacter (**B**), Carnobacterium (**D**), Butyricimonas (**F**), Alloprevotella (**G**), Agrobacterium (**H**), as well as Rhodobacter (**I**), Millionella (**J**), and Ruminococcus (**K**) were highly represented in the ND group than those in the BE group. On the other side, Cupriavidus (**C**) and Alloprevotella (**G**) were more abundant in the ND than in the E group, while Fretibacterium (**L**) was the opposite. Furthermore, the relative abundance of Enorma (**E**) was significantly higher in the BE group compared to the BB group. “*” indicated significant difference between the two groups (q value < 0.05), “**” indicated significant difference between the two groups (q value < 0.01). ND: normal diet + purified water; HFD: high-fat diet + purified water; E: high-fat diet +100 mg/kg·bw EGCG; BB: high-fat diet + 100 mg/kg·bw β-CD; BE: high-fat diet + 200 mg/kg·bw EGCG@β-CD NPs (100 mg/kg·bw EGCG).

**Figure 7 molecules-27-04894-f007:**
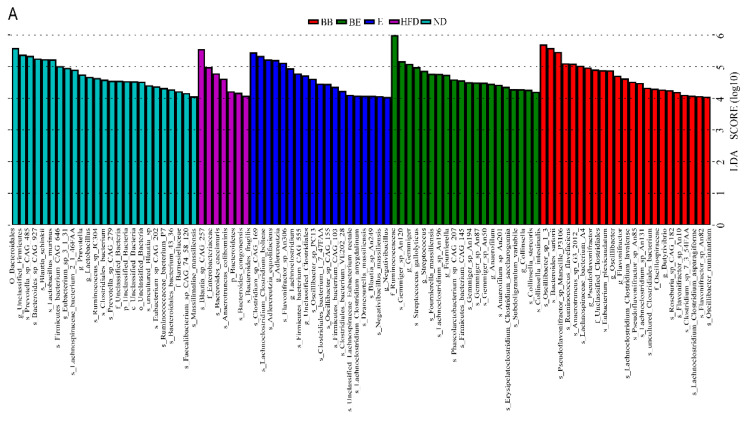
LEfSe analysis of gut microbiota among the samples. (**A**) The LDA value distribution histogram represented species with significant differences between groups, and the length of the histogram showed the size of the influence of the different species (the default setting is 4). (**B**) LEfSe analyzed the most significant abundant taxa in gut microbiota of the high-fat-diet-induced hyperlipidemia rat model. BB-enriched taxa (Red); BE-enriched taxa (Green); E-enriched taxa (Blue); HFD-enriched taxa (Purple); ND-enriched taxa (Palegreen). The diameter of the small circle was directly proportional to the relative abundance. ND: normal diet + purified water; HFD: high-fat diet + purified water; SIM: high-fat diet + 5 mg/kg·bw simvastatin; E: high-fat diet + 100 mg/kg·bw EGCG; BB: high-fat diet + 100 mg/kg·bw β-CD; BE: high-fat diet + 200 mg/kg·bw EGCG@β-CD NPs (100 mg/kg·bw EGCG).

**Table 1 molecules-27-04894-t001:** Comparative analysis in lipid accumulation and peroxidation of rats in different groups after 8 weeks.

Parameters	ND	HFD	SIM	BB	E	BE
Serum levels						
TC (mmol/L)	3.71 ± 0.38 ^c^	6.42 ± 0.64 ^a^	4.35 ± 0.51 ^c^	5.88 ± 0.42 ^ab^	5.33 ± 0.57 ^b^	4.65 ± 0.51 ^bc^
LDL-C (mmol/L)	0.32 ± 0.126 ^b^	0.52 ± 0.07 ^a^	0.40 ± 0.03 ^b^	0.50 ± 0.05 ^ab^	0.45 ± 0.08 ^ab^	0.41 ± 0.03 ^b^
HDL-C (mmol/L)	0.68 ± 0.17	0.60 ± 0.26	0.65 ± 0.12	0.63 ± 0.39	0.70 ± 0.37	0.72 ± 0.26
PL (ng/mL)	9.2 ± 0.85 ^c^	13.83 ± 1.30 ^a^	9.9 ± 1.26 ^c^	12.34 ± 0.81 ^ab^	11.83 ± 1.03 ^b^	11.07 ± 1.17 ^bc^
apo-A1 (ng/mL)	8.26 ± 0.54 ^a^	4.96 ± 0.61 ^c^	7.67 ± 0.70 ^ab^	6.09 ± 0.58 ^bc^	5.75 ± 0.70 ^bc^	6.19 ± 0.92 ^b^
Liver levels						
TC (mmol/g prot)	4.72 ± 0.76 ^c^	8.29 ± 0.72 ^a^	6.67 ± 1.19 ^b^	8.67 ± 0.93 ^a^	7.38 ± 1.37 ^ab^	5.96 ± 0.91 ^bc^
TG (nmol/g prot)	12.43 ± 1.68 ^c^	20.13 ± 4.94 ^b^	18.28 ± 2.33 ^b^	25.57 ± 1.88 ^a^	16.43 ± 1.86 ^bc^	11.00 ± 4.74 ^c^
MDA (nmol/mg)	0.64 ± 0.18 ^b^	1.32 ± 0.40 ^a^	0.81 ± 0.09 ^b^	1.03 ± 0.22 ^ab^	0.97 ± 0.22 ^b^	0.89 ± 0.10 ^b^
SOD (U/mg)	907.55 ± 65.04 ^a^	746.99 ± 36.24 ^b^	771.14 ± 90.13 ^b^	778.76 ± 110.84 ^b^	788.23 ± 40.75 ^b^	796.93 ± 60.36 ^b^
CAT (U/mg)	55.06 ± 7.62 ^a^	32.05 ± 4.95 ^b^	37.72 ± 8.10 ^b^	34.94 ± 7.00 ^b^	39.01 ± 9.87 ^b^	48.57 ± 2.66 ^ab^
GSH-Px (U/mg)	726.47 ± 156.15 ^a^	537.79 ± 57.00 ^b^	601.05 ± 124.67 ^ab^	564.52 ± 127.25 ^ab^	588.75 ± 127.44 ^ab^	663.50 ± 62.25 ^ab^

ND: normal diet + purified water; HFD: high-fat diet + purified water; SIM: high-fat diet +5 mg/kg·bw simvastatin; BB: high-fat diet +100 mg/kg·bw β-CD; E: high-fat diet +100 mg/kg·bw EGCG; BE: high-fat diet +200 mg/kg·bw EGCG@β-CD NPs (100 mg/kg·bw EGCG); TC: cholesterol; LDL-C: low density lipoprotein cholesterol; HDL-C: high density lipoprotein cholesterol; PL: phospholipids; apo-A1: apolipoprotein-A1; MDA: malondialdehyde; SOD: superoxide dismutase; CAT: catalase; GSH-Px: glutathione peroxidase. The results are expressed as mean ± *SE* (*n* = 8). Superscript characters within a row are significantly different (*p* < 0.05).

## Data Availability

Not applicable.

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
