# Peer review of "Metagenomics Approach to the Intestinal Microbiome Structure and Abundance in High-Fat-Diet-Induced Hyperlipidemic Rat Fed with (−)-Epigallocatechin-3-Gallate Nanoparticles"

_molecules, 2022, doi:10.3390/molecules27154894_

Round 1

Reviewer 1 Report

This is a very interesting study concerning the influence of (-)-Epigallocatechin-3-gallate Nanoparticles on the intestinal microbiome structure

This is a relevant study the results support the conclusion and the experiments are well designed.

However, there are minor changes that would enhance publication quality.

1)            Page 1 Line 17: The sentence “Oral administration of nanoparticles (NPs) increases the likelihood, which can encounter microorganisms in the gastrointestinal tract” does not make sense.

2)            Page 2 Line 62 and 63: “in vivo” should be in italic.

3)            Page 3, Line 100: Please explain what EE stands for. Maybe in the title 2.2. Characterisation of EGCG@β-CD NPs on embedding effect (EE).

4)            Please explain all abbreviations when they appear the first time. (eg.: PDI, SEM, XRD, FT-IR)

This is a very important study that should be revised so that its publication is possible.

Author Response

Dear Reviewer,

Thank you very much for your comments and suggestions for providing revised version of
my manuscript.

The manuscript has been modified according to the requirements of the reviewers, please see the attachment for details.

Thank you once again. We are looking forward to hearing from you.

Best regards,

Zhiyin CHEN

Associate professor, doctor

E-Mail: [email protected]

Reviewer 2 Report

The submitted manuscript is well designed and written. There are a lot of abbreviations in the work that make reading very difficult. And for some I am not even sure that they are explained. The same applies to figures.  In addition, the manufacturer's names for (-)-epigallocatechin-3-gallate (EGCG) and β-cyclodextrin should be added.

Author Response

Dear Reviewer,

Thank you very much for your comments and suggestions for providing revised version of my manuscript.

The manuscript has been modified according to the requirements, please see the attachment for details.

Thank you once again. We are looking forward to hearing from you.

Best regards,

Zhiyin CHEN

Associate professor, doctor

E-Mail: [email protected]

Reviewer 3 Report

The experiment conducted by the authors is an outstanding contribution to the field of Nanotechnological applications in gut health promotion and different metabolic disorders.

As per the objectives; all the data are well furnished and described. This manuscript must publish in a reputed journal like Molecules.

If possible Authors can take my suggestion as granted to give a more clear idea:

MD simulation of EGCG  with β-CD gives a very transparent binding potential. Authors can do at least at 100ns frequency.

Some Spelling mistakes and grammar issues are there, but they can be solved by final revision and language editing.

Author Response

Dear Reviewer,

Thank you very much for your comments and suggestions for providing revised version of my manuscript.

The manuscript has been modified according to your requirements, please see the attachment for details.

Point 1: MD simulation of EGCG with β-CD gives a very transparent binding potential. Authors can do at least at 100ns frequency.

Response 1: Thank you for your good advice, we will continue to work hard to further reduce the particle size.

Point 2: Some Spelling mistakes and grammar issues are there, but they can be solved by final revision and language editing.

Response 2: I checked the manuscript again and again, and made changes to the following content, as follows: Page 2 Line 92: “An round bottom flask” has been changed to ““A round bottom flask”; Page 4 Line 172:” Abmole Bioscience Inc” has been changed to “Abmole Bioscience Inc.”; Page 4 Line 189: “malondialde-hyde” has been changed to “malondialdehyde”; Page 13 Line 425: ”a opposite tendency” has been changed to ” an opposite tendency”.

Thank you once again. We are looking forward to hearing from you.

Best regards,

Zhiyin CHEN

Associate professor, doctor

E-Mail: [email protected]